# Soil Water Movement Changes Associated with Revegetation on the Loess Plateau of China

**Haocheng Ke [1], Peng Li [1],*, Zhanbin Li [1,2], Peng Shi [1] and Jingming Hou [1]**

[1]  State Key Laboratory of Eco-hydraulics in Northwest Arid Region of China, Xi'an University of Technology, Xi'an 710048, China; haockegsau@163.com (H.K.); zhanbinli@126.com (Z.L.); shipeng015@163.com (P.S.); jingming.hou@xaut.edu.cn (J.H.)

[2]  State Key Laboratory of Soil Erosion and Dry Land Farming on the Loess Plateau, Institute of Soil and Water Conservation, Chinese Academy of Sciences and Ministry of Water Resources, Yangling 712100, China

*  Correspondence: lipeng74@163.com; Tel.: +86-29-82312658

**Abstract:** Soil water is the limitation factors in the semiarid region for vegetation growth. With the large scale "Grain for Green" implementation on the Loess Plateau of China, an amount of sloping cropland was converted to forestland, shrubland, and grassland. The spatial and temporal distribution of soil water was changed. However, the effect of revegetation on soil water movement is still unclear. In this study, we analyze the stable isotopes changes in precipitation and soil water in sloping cropland, forestland, shrubland, and grassland to trace the movement of moisture in soil. The results showed that $\delta^{18}O$ in shallow layers (<20 cm depth) of sloping cropland, forestland, shrubland, and grassland were $-3.54‰$, $-2.68‰$, $-4.00‰$, and $-3.16‰$, respectively. The $\delta^{18}O$ in these layers were higher than that in the lower layers, indicating that evaporation was mainly from the shallow layers. The $\delta^{18}O$ for the soil water in the unsaturated zone in the grassland, shrubland, and forestland of the temporal variability decreases with depth and approaches a minimum value at 160 cm, 180 cm, and 200 cm, respectively, suggesting that the soil water is relatively stable many months or even longer. Precipitation was infiltrated with piston and preferential modes, and infiltration demonstrated obvious mixing. Present study demonstrated the $\delta^{18}O$ was more sensitive than the soil water content for tracing the maximum infiltration depth of event water and recharge mechanisms. Consequently, we suggested that the land user management such as type, plant density should be considered in the revegetation.

**Keywords:** the Loess Plateau of China; oxygen isotope; precipitation; soil water; unsaturated zone

## 1. Introduction

Acting as a limitation factors for the vegetation growth, soil water is crucial to the hydrological cycles in semiarid environments on the Loess Plateau of China (LPC). The Loess Plateau has the greatest erosion rate in the world. One of the causes of soil erosion is excessive use of land [1,2]. With the large-scale "Grain for Green" implementation on the Loess Plateau, many of the sloping croplands were converted to forestland, shrubland, and grassland. The world is trying to achieve the sustainable goals for development by the United Nations and achieve the land degradation neutrality via the use of restoration [3,4]. Land use type is one of the main driving forces affecting the evolution of soil and water resources in unsaturated zones [5–8]. Understanding the movement of soil water in unsaturated zones in different land use types is important for understanding the distribution and cycling of water, salt, and other nutrients, and for improving land management for controlling erosion and runoff in revegetation and ecological construction [9,10]. However, the influence of land use for the soil water movement through unsaturated zones on the Loess Plateau has been reported in a few papers. Accordingly, research on

soil water with land use types play an important role in the mechanism of soil water movement and soil water supply system on the Loess Plateau. The key research of ecohydrological started to delve into the relationship between soil water and subsurface mixing, as well as its interaction and feedback with the ecosystem [11–13]. Little attention, however, has been paid to the effect of revegetation on soil water movement in unsaturated zones on the Loess Plateau.

The stable hydrogen and oxygen isotopes of soil water have been used to study the infiltration, evaporation, and mixing processes and quantitatively evaluates the groundwater recharge and evaporation rates, which is hard to get by other technologies [14–16]. As isotope ratios of oxygen and hydrogen in soil water impact these ratios differently, they can be used to distinguish between transpiration and evaporation. The evaporation enriches the soil water of oxygen and hydrogen isotopes, and the transpiration process does not fractionate isotopes [17–20]. Many early studies have reported oxygen isotope- and hydrogen isotope-enrichment in soil water near the surface [21,22]. A seminal study by Landwehr and Coplen developed a new method that is helpful in deriving isotope fractionation due to soil evaporation [23]. A review on groundwater recharge estimates via stable isotope of the unsaturated zone was provided by Koeniger et al. [24]. The movement of soil water in unsaturated zone under various climate conditions, inclusive of arid and semiarid, has been studied to contrast movement in various processes such as the infiltration of precipitation, mixing, and evapotranspiration in a profile of soil [25]. The stable hydrogen and oxygen isotope have been used for the identification of spring and stream water sources in high mountain regions, for the identification of flow properties, and hydrodynamic parameters of springs in karstic environments [26–29]. The stable isotopes have also been identified for the contribution of rainfall to groundwater discharge in San Vittorino Plain, for the identification of processes, structure, and status of the groundwater system [30,31]. The flow of soil water in unsaturated zones, however, is complex and affected by vegetation cover, wetness, structure, and soil texture. Infiltration of precipitation and subsequent downward percolation are described to be piston and preferential flow [32–34]. For piston flow, the soil water produced from the recent precipitation will reduce the amount of residual soil water that is older, and is normally combined with it [35]. Preferential flows occur in unsaturated zones through macropores attributed to forces, inclusive of earthworm burrows, decayed plant roots, and cracks [36,37]. Different flow mechanisms result in different isotope profile. The use of oxygen and hydrogen stable isotopic compositions for tracing the soil water movement in unsaturated zones and to determine the characteristics of infiltration in a soil profile, however, has rarely been applied on the Loess Plateau under revegetation and ecological construction as a function of time.

We examined the seasonal variations in stable oxygen isotope of soil water in unsaturated zones and precipitation for four typical land uses on the Loess Plateau to identify the factors controlling soil water movement within the unsaturated zone. The objectives of the present study are: (1) To determine the temporal and spatial variations of soil water content; (2) to analyze stable oxygen isotopes of the precipitation and differences in profile soil water $\delta^{18}O$ for four land use types under revegetation and ecological construction; and (3) to study the infiltration mechanisms and influence factors in unsaturated zones based on $\delta^{18}O$ value characteristics in the study area.

## 2. Materials and Methods

### 2.1. Study Area

The Wangmaogou watershed (37°34′13″–37°36′03″ N, 110°20′26″–110°22′46″ E) is in the central region of the Loess Plateau in Northern Shaanxi Province in China. The watershed has an area of 5.97 km² (Figure 1) and an altitude between 940 and 1188 m a.s.l. The region has a semiarid continental climate with a mean annual temperature of 10.2 °C. Monthly mean temperatures range from 23.5 °C in July to −7.5 °C in January (Figure 2a). The mean annual precipitation is 513 mm with high interannual variability, and 70% falls between June and September. The ravine density in the watershed is 4.3 km km⁻², with interfluves accounting for 53.3% and valleys accounting for 46.7%. The watershed is a

typical loessial hilly and gully landform. The difference in altitude from top to bottom is typically from 100 to 200 m. The soil of the study area has developed on wind-deposited loessial parental material [38]. The most common soil is loess with silt contents range from 62% to 73% and clay contents range from 17% to 20% (Table 1). The Wangmaogou watershed responds to the country policy of Grain to Green Project (GTGP) since 1953. With the developing of GTGP on the Loess Plateau, many of the sloping croplands were converted to forestland, shrubland, and grassland.

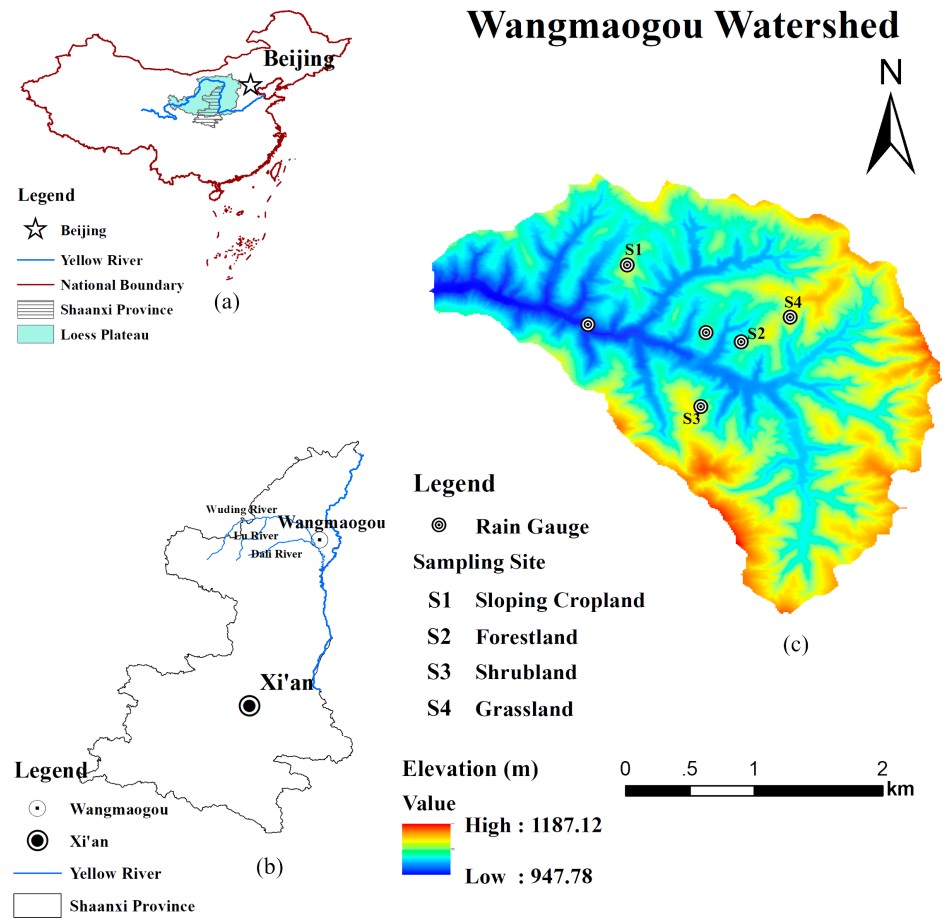

**Figure 1.** Location of the study area in Shaanxi Province, China (**a**,**b**); sampling sites (**c**).

**Table 1.** Soil particle-size distribution and bulk density in the six land uses.

| Land Use Types | Sand (2–0.05 mm) (%) | Silt (0.05–0.002 mm) (%) | Clay (<0.002 mm) (%) | Sand: Clay Ratio | Bulk Density (g cm$^{-3}$) |
|---|---|---|---|---|---|
| Grassland | 16.19 | 64.78 | 19.03 | 0.85 | 1.36 |
| Shrubland | 13.54 | 69.13 | 17.33 | 0.78 | 1.38 |
| Forestland | 10.63 | 70.79 | 18.58 | 0.57 | 1.4 |
| Sloping cropland | 14.65 | 65.62 | 19.73 | 0.74 | 1.21 |

Current land uses investigated were grassland, shrubland, forestland, and sloping cropland (Table 2).

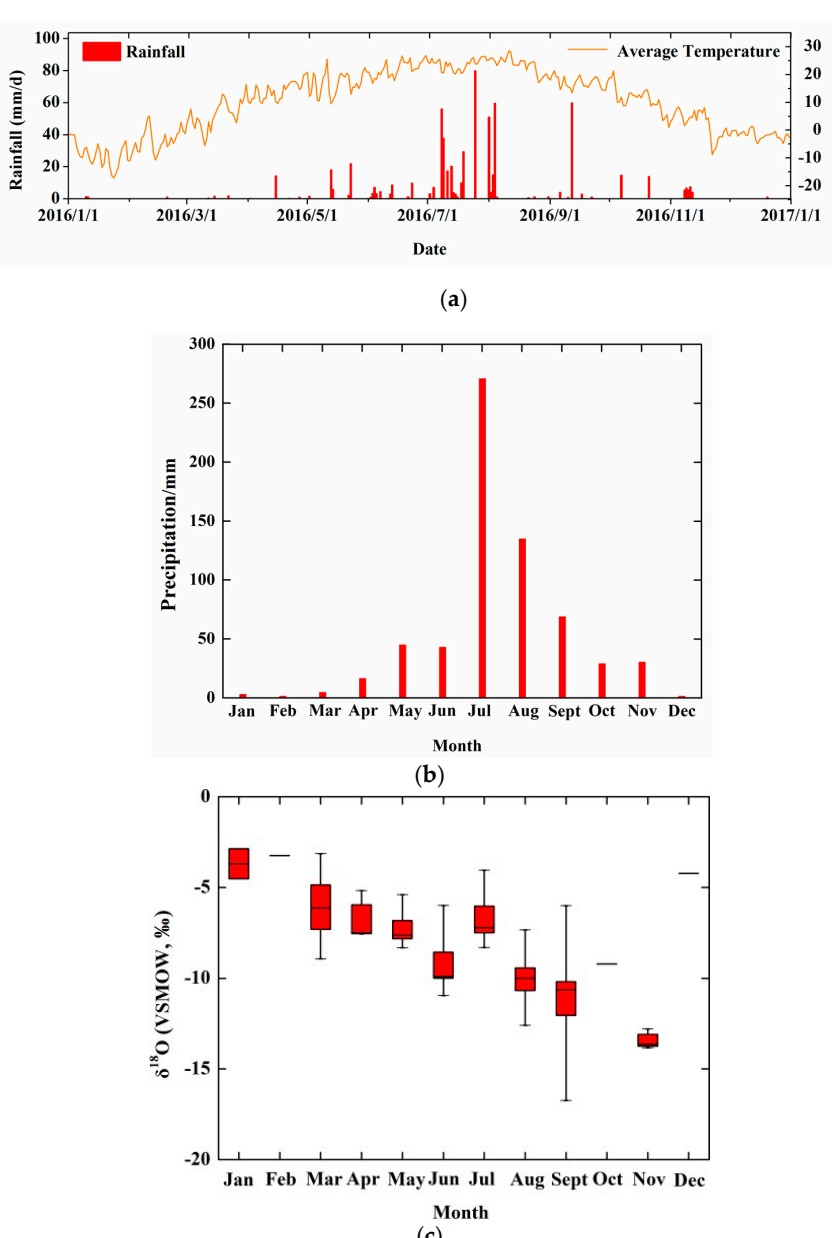

**Figure 2.** Daily variation of (**a**) temperature and precipitation and monthly variation of precipitation amount (**b**) and precipitation δ$^{18}$O (**c**) at the study site.

**Table 2.** Site description.

| Land Use Types | Altitude (m) | Location | Soil Texture | Main Vegetation | Vegetation Coverage (%) |
|---|---|---|---|---|---|
| Grassland | 988.56 | 110°21′40.263″ E 37°35′21.352″ N | Loessial soil | *Bothriochloa ischaemum* | 85–90 |
| Shrubland | 1006.36 | 110°21′55.923″ E 37°35′17.696″ N | Loessial soil | *Caragana korshinskii* | 80–90 |
| Forestland | 1009.45 | 110°21′53.172″ E 37°34′50.624″ N | Loessial soil | *Pinus tabuliformis* | 75–80 |
| Sloping cropland | 968.39 | 110°21′11.708″ E 37°34′51.673″ N | Loessial soil | *Solanum tuberosum* | 80–85 |

*2.2. Methods*

2.2.1. Sampling Methods

Under comprehensive consideration of land use types of revegetation, soil samples were collected from the grassland, shrubland, forestland, and sloping cropland in May, August, and November 2016 by drilling with a hand auger of 9 cm in diameter, at 10 cm intervals to a depth of 100 cm and at 20 cm intervals from 100 to 400 cm. Three replicates were collected for each soil sample. The parts of soil samples were packed into an aluminum box sealed with plastic wrap and measured for gravimetric soil water content. The parts of soil samples were then packed into glass bottles sealed with Parafilm to prevent evaporation and promptly brought to State Key Laboratory of Eco-hydraulics in Northwest Arid Region of China, Xi'an University of Technology, stored at −4 °C until laboratory analysis. A Mastersizer 2000 laser particle size analyzer (Malvern Instruments, Malvern, UK) was used to measure the percentages by volume of the particle sizes. Four undisturbed soil samples per replicate treatment were randomly selected from the soil layer (0–10 cm) for laboratory experiments using 50 mm × 54 mm cylindrical cores. Soil bulk density was determined from oven-dried undisturbed cores as mass per volume of oven-dried soil. Samples of precipitation were collected from the six sampling sites from January to December 2016. This work collected the precipitation using the "Rain Gauge" available in an all weather, into which a small amount ($\approx$ 3 mL) of mineral oil had been added to prevent evaporation. During the winter months, snow was collected in the same collector and melted to determine the water equivalent. A total of 50 precipitation samples were collected from each sampling site. Three replicates were collected for each precipitation sample. Samples of precipitation were collected in glass bottles sealed with Parafilm to prevent evaporation and promptly brought to State Key Laboratory of Eco-hydraulics in Northwest Arid Region of China, Xi'an University of Technology, stored at −4 °C until laboratory analysis.

2.2.2. Gravimetric Soil Water Content

The content of soil water was measured by oven-drying. The gravimetric soil water content (SWC) of each sample was measured by weighing the samples in the field, heating the sample for 24 h at 100 °C, and weighing the dry soil.

$$\text{SWC(\%)} = [(M_W - M_D)/M_D] \times 100 \qquad (1)$$

where SWC is the gravimetric soil water content (%), $M_w$ is the weight of the samples in the field (g), and $M_D$ is the weight of the dry soil (g).

2.2.3. Measurements of Hydrogen and Oxygen Isotopes

A LI-2000 (LICA, China) liquid water vacuum extraction method was used to extract the soil water. The stable hydrogen and oxygen isotope composition ($\delta^{18}O$ and $\delta^2H$) of the soil water was determined using an LGR liquid water isotope analyzer (Los Gatos Research Inc., USA) at the State Key Laboratory of Eco-hydraulics in Northwest Arid Region of China, Xi'an University of Technology.

Each sample was injected six times, but the first two injections were discarded to eliminate cross contamination. The accuracy of the measurements was ±0.3‰ for $\delta^2H$ and ±0.1‰ for $\delta^{18}O$. The sample-injection volume ($2.50 \times 10^{16}$ to $4.50 \times 10^{16}$ water molecules, ±3.00%), temperature (±1 °C), and the accuracy of the $\delta^2H$ and $\delta^{18}O$ measurements were checked. The test results that did not satisfy these conditions were eliminated from the analysis. The results are expressed as δ values relative to Vienna standard mean ocean water (VSMOW) in per mil (‰):

$$\delta_{\text{sample}}(\%0) = \left( R_{\text{sample}}/R_{\text{VSMOW}} - 1 \right) \times 10^3 \qquad (2)$$

where $\delta_{sample}$ is the deviation of the isotope ratio of a sample relative from that of VSMOW; $R_{sample}$ is the ratio of $^2$H to $^1$H (or $^{18}$O to $^{16}$O) in the sample; and $R_{VSMOW}$ is the ratio of $^2$H to $^1$H (or $^{18}$O to $^{16}$O) in VSMOW.

### 2.2.4. Source Partitioning Using Stable Isotopes

The isotopic mass balance was calculated using Equations (3) and (4) [39]:

$$\delta_{mix} = f_{old}\delta_{old} + f_{new}\delta_{new} \tag{3}$$

$$f_{new} + f_{old} = 1 \tag{4}$$

where $f_{new}$ and $f_{old}$ are the proportions of new and old water ($f_{new} + f_{old} = 1$), respectively; $\delta_{new}$ and $\delta_{old}$ are the isotopic signatures of the new and old water, respectively, and $\delta_{mix}$ is the isotopic signature of the mixed water.

### 2.2.5. Calculation of Biomass

Tree biomass was calculated by:

$$LnW_1 = 0.9128LnD^2H - 3.0156 \left( R^2 = 0.93 \right) \tag{5}$$

$$LnW_2 = 0.9088LnD^2H - 3.1683 \left( R^2 = 0.94 \right) \tag{6}$$

$$W_{Ti} = W_1 + W_2 \tag{7}$$

where H is the tree height, D denotes the diameter at breast height, and $W_1$ and $W_2$ are the above- and belowground biomasses, respectively [40,41]. Samples of rootlet biomass from herbaceous plants were collected by drilling (inner diameter of 100 mm) at 10-cm intervals to a depth of 100 cm, and every horizontal was mixed in a bag. The soil was then soaked and then washed under the tap water with a 0.25 mm sieve to isolate plant roots. The roots were air-dried, weighed, and stored at 65 °C to calculate the biomass.

### 2.2.6. Statistical Analysis

All statistical analyses were performed using SPSS 19.0 and Origin 8.5. One-way ANOVAs followed by Tukey's HSD tests ($p < 0.05$) were used to compare the effect of seasonal changes and land use types on SWC.

## 3. Results

### 3.1. Isotopic Composition of the Precipitation

The statistical characteristics of the precipitation isotopes are shown in Figure 2. The $\delta^{18}$O of the precipitation varied greatly during the year. The precipitation isotopes from June to November were more depleted and were enriched in the study area from December to May (Figure 2).

### 3.2. Profile Variations of Soil Water in the Unsaturated Zone in the Four Land Uses

Figure 3 provides the profile variations of soil water in the unsaturated zone for the four land uses. The SWC ranged from 4.74% to 25.94%. The SWC in May was as follows: Sloping cropland (16.59%) > grassland (14.49%) > forestland (11.82%) > shrubland (10.57%). The SWC in August was as follows: Sloping cropland (17.38%) > shrubland (13.49%) > grassland (9.94%) > forestland (8.92%). The SWC in November was as follows: Sloping cropland (16.82%) > shrubland (13.65%) > forestland (12.67%) > grassland (12.64%).

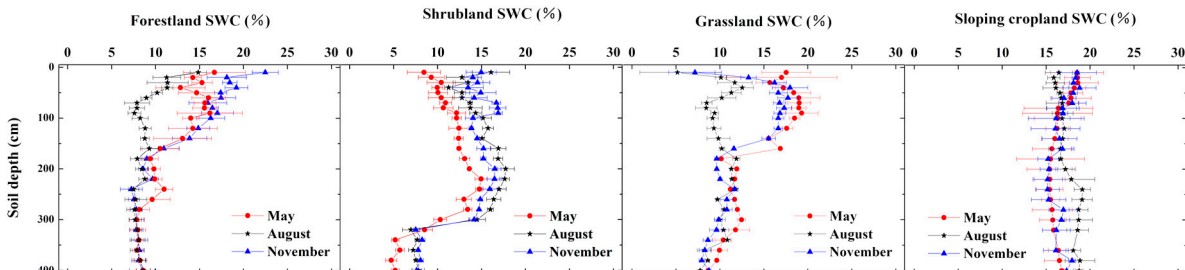

**Figure 3.** Distribution of soil water content (SWC) in the four land use types. The standard deviation represents the dispersion degree of the data.

The results shown in Figure 3 indicate that the SWC located in the shallow layers presents a big difference, but the SWC in the deeper layers was most stable for grassland, forestland, and shrubland. Sloping cropland retained water the best, with the highest SWCs throughout the study period. SWC in the 40–100 cm layers show a different value with the change of growing season.

### 3.3. Biomass of Four Land Use Types

Table 3 shows the vegetation cover's root distributions in the four land uses. Mean maize rootlet biomass in the upper 30 cm of the sloping cropland accounted for 84% of the total biomass. The root distribution of *Bothriochloa ischaemum* was well-proportioned, except for the lowest values in the 80–100 cm layer (Figure 3). The roots of *Pinus tabuliformis* were concentrated in the 40–60 cm layers, corresponding to the peaks of water loss in the profiles.

**Table 3.** Biomasses in the six land uses.

| Depth (cm) | Grassland (kg m$^{-2}$) | Shrubland (kg m$^{-2}$) | Forestland (kg tree$^{-1}$) | Sloping Cropland (kg m$^{-2}$) |
|---|---|---|---|---|
| Aboveground biomass | 0.82 | 1.95 | 185.5 | 1.85 |
| 0–10 | 0.16 | 0.12 | 4.31 | 0.1 |
| 10–20 | 0.25 | 0.24 | 6.65 | 0.27 |
| 20–30 | 0.15 | 0.29 | 10.65 | 0.29 |
| 30–40 | 0.11 | 0.43 | 14.23 | 0.08 |
| 40–50 | 0.1 | 0.25 | 42.08 | 0.01 |
| 50–60 | 0.08 | 0.23 | 20.87 | 0.01 |
| 60–70 | 0.04 | 0.18 | 19.01 | 0.01 |
| 70–80 | 0.03 | 0.16 | 16.97 | 0 |
| 80–90 | 0.01 | 0.08 | 18.93 | 0 |
| 90–100 | 0 | 0.04 | 10.93 | 0 |
| Belowground Biomass | 0.93 | 2.02 | 164.63 | 0.77 |

### 3.4. Profile Variations of $\delta^{18}O$ in the Four Land Use Types

Figures 4 and 5 show the characteristics of $\delta^{18}O$ for the soil water in the unsaturated zone in the four land uses in May, August, and November. The $\delta^{18}O$ in May ranged from −10.77‰ to 4.85‰, with a mean of −7.61‰, and averaged −7.67‰ in the grassland, −7.76‰ in the shrubland, −7.83‰ in the forestland, and −7.19‰ in the sloping cropland. The $\delta^{18}O$ in August ranged from −10.65‰ to 0.10‰, with a mean of −8.24‰, and averaged −8.33‰ in the grassland, −8.12‰ in the shrubland, −8.14‰ in the forestland, and −8.35‰ in the sloping cropland. The $\delta^{18}O$ in November ranged from −13.23 to −2.85‰, with a mean of −9.14‰, and averaged −9.02‰ in the grassland, −9.33‰ in the shrubland, −8.55‰ in the forestland, and −9.65‰ in the sloping cropland. The decreasing trend of $\delta^{18}O$ was interrupted by precipitation early in August and November and peaked again in November in the 80–90 cm layer. The $\delta^{18}O$ was stable in deep soil (>200 cm) and varied seasonally the most in the 0–10 cm layer. The $\delta^{18}O$ decreased significantly with depth and stabilized in the deep layers.

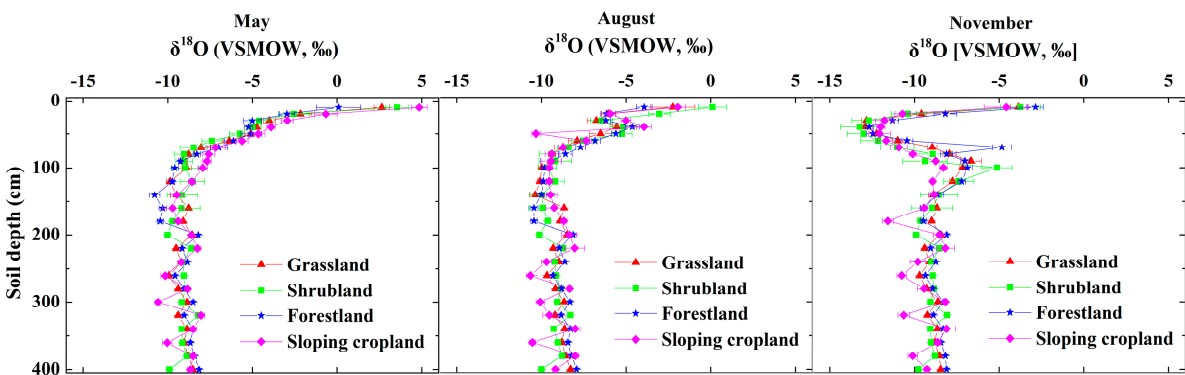

**Figure 4.** Variation of soil-water $\delta^{18}$O in the profiles of the four land use types.

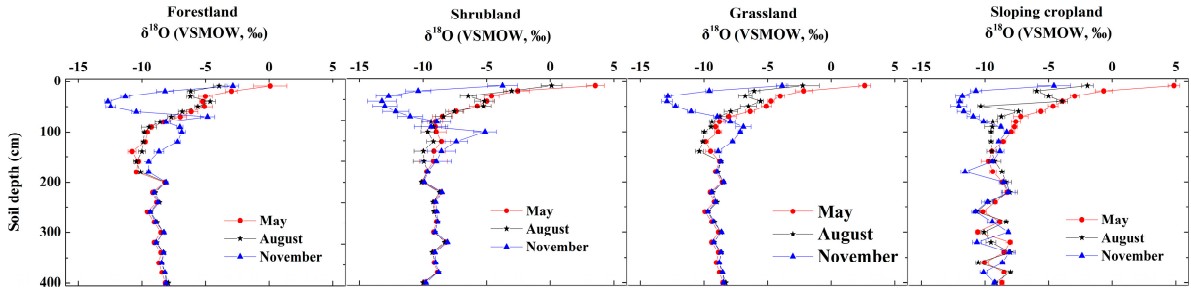

**Figure 5.** Seasonal variation of soil-water $\delta^{18}$O in the profiles of the four land use types.

The source of the soil water became depleted in 18-oxygen from May to November. The peak negative value was recorded at depths of 40 cm in the grassland and forestland, 50 cm in the shrubland and sloping cropland; rain fell on 8 and 12 November 2016 (30 mm). The $\delta^{18}$O of the soil water in the 0–10 cm layer in the four land use types indicated 18-oxygen enrichment, and 18-oxygen was less abundant in the deepest layers. The $\delta^{18}$O was relatively stable in the layers below 200 cm and varied little seasonally. Soil water $\delta^{18}$O ranged from $-9.93‰$ to $-8.28‰$ in the grassland, ranged from $-10.11‰$ to $-8.07‰$ in the shrubland, and ranged from $-9.57‰$ to $-7.92‰$ in the forestland under 160 cm, 180 cm, and 200 cm, respectively.

### 3.5. Composition of $\delta^{18}$O and $\delta^{2}$H in the Four Land Use Types

The Local Meteoric Water Line (LMWL) is given in Figure 6. The Local Meteoric Water Line (LMWL, Figure 6) was deviation to the Global Meteoric Water Line (GMWL) ($\delta^{2}$H = 8$\delta^{18}$O + 10). The values of $\delta^{18}$O varied between $-16.74$ and $-2.87‰$ with a mean of $-8.54‰$. The $\delta^{18}$O for a storm (60 mm) on 12 September 2016 was $-15.19‰$. The $\delta^{18}$O for a larger storm (80 mm) on 25 July was $-8.31‰$. Figure 6 show the soil water hydrogen and oxygen stable isotopic composition in the unsaturated zone in the four land use types in May, August, and November. Cross-plotting the 2-hydrogen and 18-oxygen isotopic records with different slopes suggests seasonally variable evaporative enrichment of 18-oxygen (the red, olive, and blue lines represent the isotopic profiles for May, August, and November, respectively). The slope of the regression line for soil water evaporation differed among the four land use types. The effect of soil evaporation in May distribution from big to small was as follows: Sloping cropland, shrubland, grassland, and forestland. For August, the list was sloping cropland, shrubland, grassland, and forestland. Meanwhile, the evaporation effect of soil in November obey the following distribution: Sloping cropland, grassland, shrubland, forestland.

The isotopic mass balance was calculated using Equations (3) and (4). Mixing occurred along all profiles, with $f_{new}$ averaging 74%, 73%, 64%, and 68% in the grassland, shrubland, forestland, and sloping cropland, respectively, above 100 cm after continuous precipitation (130 mm).

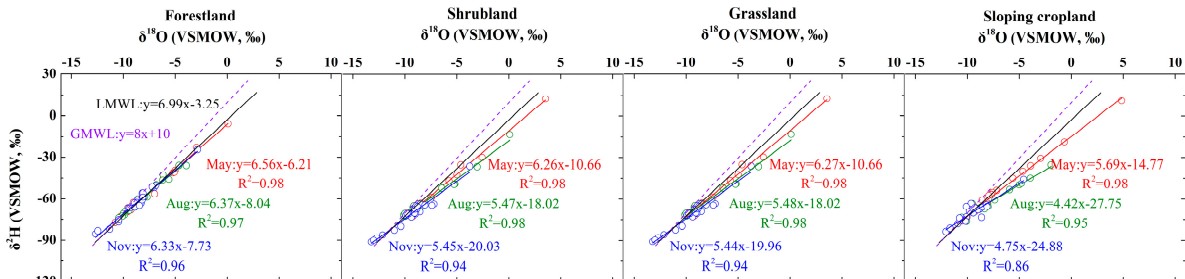

**Figure 6.** The soil water isotopic composition of the four land use types. The black lines represent the global meteoric water line (GMWL) of Carig (1961) shown for reference. The violet lines represent the local meteoric water line (LMWL).

## 4. Discussion

### 4.1. Features of the Precipitation Isotopes

Combining the isotopic values for precipitation from January to December 2016 (Figures 2c and 6), the local meteoric water Lline (LMWL) were given in Figure 6. The precipitation isotopes from June to November were more depleted and were enriched in the study area from December to May, maybe because the precipitation from June to November came from maritime air masses, and the precipitation from December to May was predominantly from continental air masses [42,43]. The LMWL from the study site deviated from to the GMWL, which is attributed to enrichment due to evaporation (Figure 6). The precipitation isotopes in our study varied seasonally and were strongly affected by rainfall (Figure 2). Heavy rainfall generally depleted the isotopic compositions. For example, $\delta^{18}O$ for a storm (60 mm) on 12 September 2016 was $-15.19‰$. The $\delta^{18}O$ for a larger storm (80 mm) on 25 July, however, was $-8.31‰$. The $\delta^{18}O$ for this sample therefore represented $\delta^{18}O$ for the initial rainfall carrying abundant heavy isotopes, other than those of the whole rainfall.

### 4.2. Effect of Evaporation on Soil Water

In our study area, evaporation is a major factor for the enrichment of 18-oxygen of soil water in shallow layers. A comparison of the regression lines for soil water with GMWL and LMWL, specifically $\delta^2H = 8\delta^{18}O + 10$ [44] and $\delta^2H = 6.99\delta^{18}O - 3.25$ ($R^2 = 0.96$, n = 50), indicated that the effect of evaporation varied among the land use types (Figure 6). The regression lines from four land use types deviated from the LMWL, which arises from enrichment of 18-oxygen of soil water because of evaporation. The $\delta^{18}O$ differentially varied with each season for the four land use types (Figure 5). The non-equilibrium fractionation of evaporation made 18-oxygen of soil water enrichment around the soil surface. The isotopic profiles for May, August, and November indicated that most of the variation in $\delta^{18}O$ was in the upper 200 cm for the four land use types, due to evaporation, infiltration, as well as mixing. Dry conditions caused rainwater to strongly evaporate, and confined the SWC recharge, as suggested by the negative peaks in May and August. The $\delta^{18}O$ of soil water in the unsaturated zone is also affected by precipitation. So, the decline in the soil water isotopic profile is not as gradual as Zimmerman predicted [14]. The $\delta^{18}O$ varied slightly seasonally under the 200 cm soil layers (Figures 4 and 5), indicating that evaporation had a weak effect on deep soil water. Evaporation caused 18-oxygen enrichment in soil water around the surface, which created $\delta^{18}O$ profiles that decreased in the 0–100 cm layers. The land use types affected $\delta^{18}O$ in the soil profile significantly; this phenomenon was caused by the differences of the soil physical and hydrological properties. Correspondingly, the mechanism of infiltration of soil water in the unsaturated zone may be affected by the land use types. An ANOVA analysis of $\delta^{18}O$ in the upper 100 cm indicated that soil water $\delta^{18}O$ values were significantly in the grassland, shrubland, forestland, and sloping cropland ($p = 0.015$, 0.01, 0.01, 0.04, respectively). The spatial and temporal differences in soil water isotopes in the unsaturated zone (Figures 4 and 5) were attributed to evaporation.

### 4.3. Infiltration Mechanisms based on $\delta^{18}O$ Isotopic Characteristics

The main scope of the $\delta^{18}O$ of soil water changed by the different land use types is located in the 0–100 cm layers. The $\delta^{18}O$ of soil water in the shrubland was −13.22‰ at a depth of 30–40 cm in November, and the precipitation $\delta^{18}O$ was −13.65‰, respectively. Obviously, the results shown in Figure 5 indicated that the $\delta^{18}O$ of soil water is consistent with the precipitation $\delta^{18}O$. The $\delta^{18}O$ of soil water in the shrubland and precipitation isotopic values in November suggested that piston flow occurred. The $\delta^{18}O$ of soil water in the 90–100 cm layer in the shrubland in November corresponded with $\delta^{18}O$ in the 20–30 cm layer in August, indicating that soil water migration along the profile existed with a lag effect. The isotopic profiles of the forestland and sloping cropland in August contained abruptly lower values at depths of 160 and 50 cm, respectively, indicating that preferential flow occurred in these land use types, perhaps due to the paths for preferential flow provided by root holes, wormholes, cracks, and fissures in the loess [19]. Due to this, macropores in the soil provided pathways for the rainwater during the percolation of soil water. Low SWC, especially during the rainy season (Figure 3), would facilitate the preferential flow [45]. In the process of rainwater infiltration, the piston flow and the preferential flow coexist, which was largely impacted by the land use type. Briefly, other than the sloping cropland, precipitation on the Loess Plateau rarely recharged the deep soil (>200 cm), and the types of infiltration and land use primarily impacted the isotope characteristics of the soil profile.

Generally, the $\delta^{18}O$ of soil water at the four land use types were more depleted below the 200 cm soil layer and suggest greater contribution of rainy season precipitation to replenishment. According to the Nielson and Bouma (1985) classification system: CV ≤ 10%, 10% < CV <100%, and CV ≥ 100% indicate low, middle, and strong variability, respectively [46]. The CV values of soil water $\delta^{18}O$ were −4.42% in the grassland, −5.60% in the shrubland, and −4.91% in the forestland. The CV value of soil water $\delta^{18}O$ belongs to weak variability in the grassland and shrubland forestland below 160 cm, 180 cm, and 200 cm, respectively, indicating that normal rains and even continuous rains rarely affected the soil water in the deeper layers, except for the sloping cropland. The CV value of soil water $\delta^{18}O$ belongs to strong variability in the sloping cropland throughout the year, indicating that it may have a large impact on groundwater recharge. The isotope depth profiles could be used to calibrate a numerical model and derive this way time-variant travel times, while the soil water isotope sampling is limited to snapshots into the processes (e.g., travel time) [47].

### 4.4. Tracing the Mixing of Soil Water Based on $\delta^{18}O$ Isotopic Characteristics

The hydrogeochemical characteristics of the soil water provided evidence of mixing. The amplitude of the seasonal cyclic variation in soil water $\delta^{18}O$ for each layer gradually decreased with depth (Figures 4 and 5), and the difference in $\delta^{18}O$ between precipitation and soil water could not account for evaporation under the climatic conditions in our study area, where temperatures were lower (13 °C) and relative humidity was higher (95%) in autumn. The $\delta^{18}O$ was most negative in the shrubland at −13.22‰, similar to −13.65‰ for the precipitation, further illustrating that the evaporation of precipitation may have been negligible in the enrichment during November. The $\delta^{18}O$ was most negative in the forestland at −12.67‰, in the grassland at −12.91‰, and more enriched to −13.65‰ for the precipitation, indicating that the difference may be because antecedent soil water was combined by rainwater. The $\delta^{18}O$ profile in November (Figure 5) probably denotes antecedent soil water in summer ("old water") and a mixture of precipitation in autumn ("new water"). The mixing of soil water probably increased slightly with depth. The $\delta^{18}O$ for the soil water in the grassland, shrubland, and forestland of the temporal variability decreases with depth and approaches a minimum value at 160 cm, 180 cm, and 200 cm, respectively, suggesting that the residence times of this water could be several months or more. Mixing can be caused by the infiltration or percolation of newer water into older soil water, hydrodynamic dispersion, or the interaction of preferential flow, among other causes. Mixing with "old water" could explain the isotopic enrichment of the precipitation in the profile and the effect of evaporation on mixing, suggesting that $\delta^{18}O$ is useful for tracing the mixing

of soil water in the unsaturated zone. Identifying the dominant factor(s) for the mixing based on the amount of water and the $\delta^{18}O$ data for the precipitation and soil water, however, is difficult.

*4.5. Tracing Soil Water Movement by Cross Plotting $\delta^2H$ and $\delta^{18}O$*

The results [24] indicated that the water absorbed by plant root would not fractionate the soil water isotopes. The absorption of water by the vegetation, however, would affect the water distribution in the soil profile and lead to changes in the patterns of soil water movement, which would affect the isotopic profile. It is suggested here that evaporation impact was changed with various seasons and diverse four land use types through the comparison of the regression line for soil water with the LMWL, in particular, $\delta^2H = 6.99(\delta^{18}O) - 3.25$ ($R^2 = 0.96$, n = 50). In May, there was a strong evaporation phenomenon in the sloping cropland. The lower precipitation (<50 mm) and the unexcavated slope value were lower, which could be expressed as 5.69. Furthermore, forestland with dense canopies and thick litter had a higher slope value of 6.56. Evapotranspiration became stronger in August in the four land use types. The SWC of forestland and grassland decreased with the decrease of root distribution. However, this phenomenon has not occurred in the sloping cropland (Table 3 and Figure 3). This finding requires further research. The root distribution of plants is closely related to SWC, and the water absorption layer is often related to the root distribution area [48]. The forestland, which contain *Pinus tabuliformis* as the dominant plants, have a high root biomass at 40–60 cm depth, which agree with the water loss in the profiles. Stolte et al. [10] reported a similar result. In November, the regression line for the soil water in the grassland, shrubland, forestland, and sloping cropland were expressed as y = 5.44x − 19.96 ($R^2 = 0.94$), y = 5.45x − 20.03 ($R^2 = 0.94$), y = 6.33x − 7.73 ($R^2 = 0.96$), and y = 4.75x − 24.88 ($R^2 = 0.86$), respectively, suggesting weak evaporation. The higher slope values and SWC indicated that the compensation effect occurs during the late growing season of the four land use types (Figures 3 and 6). Yet similar phenomena have not been observed in forestland. The reason may be that the dense of foliage and canopy reduce the infiltration rate of precipitation moisture retained on the surface. This assumption is supported by the SWC in November reaching 22.48% (Figure 3). And reference [49] indicated that the connectivity can provide a basis for the development of better measurement and modeling approach. This research can be conducive to have a further understanding about water quantifying and sediment transferring in catchment systems.

## 5. Conclusions

Oxygen isotopic compositions and mechanisms of soil water movement in the unsaturated zone were studied in four land use types in the Wangmaogou watershed on the Loess Plateau. The critical factors that controlled how water moved through, out of, and into the soil profile were identified. The strong evaporation on the Loess Plateau was likely the dominant factor accounting for 18-oxygen enrichment in the shallow layer, whereas the characteristics of precipitation infiltration in the various seasons were the critical factors controlling the profile dynamics of $\delta^{18}O$ in deeper soil. The $\delta^{18}O$ of the soil water were in a steady state in the grassland, shrubland, and forestland were 160 cm, 180 cm, and 200 cm, respectively. The composition of the lighter isotopes during infiltration suggested that large mixing with "old water" followed the infiltration. The mixing may have been due to the infiltration or percolation of "new water" into the "old water", hydrodynamic dispersion, or the interaction of preferential flow, among other causes.

The characteristics of the $\delta^{18}O$ isotopic composition profile indicated that piston flow and preferential flow coexisted in the infiltration of rainwater. The comparison of four land use types shows that the isotopic composition of soil water may vary due to local drainage and soil characteristics. The higher $\delta^{18}O$ in the drier land uses may partly attribute to the high evaporation 18-oxygen enrichment and different proportions of "new water" and "old water". This study did not address the time of "new water" and "old water" mixing residence and nature. Future studies using combinations of analytical techniques would help address these problems. To sum up, the $\delta^{18}O$ is suggested to be more sensitive than SWC for tracing the depth of precipitation infiltration and recharge mechanisms.



The results provide reliable reference information for similar hydrological studies in field-based hydrological studies.

**Author Contributions:** P.L. and H.K. conceived the main idea of the paper. P.L., Z.L., P.S., and J.H. designed and performed the experiment. H.K. wrote the manuscript and all authors contributed in improving the paper.

**Funding:** This project was supported by the National Key Research and Development Program of China (Contract No. 2016YFC0402404 and 2017 YFC 0504704) and the National Natural Science Foundation of China (Contract No. 41330858; 41601291 and 41471226).

**Acknowledgments:** The authors wish to acknowledge the members of the project team for investigation and sampling in the field.

**Conflicts of Interest:** The authors declare no conflict of interest.

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
