# Peer review of "Soil Water Movement Changes Associated with Revegetation on the Loess Plateau of China"

_water, doi:10.3390/w11040731_

Reviewer 1 Report

Oxygen isotopic compositions and mechanisms of soil water movement in the unsaturated zone were studied in four land use types in the Wangmaogou watershed on the Loess Plateau. The critical factors that controlled how water moved through, out of and into the soil profile were identified.

It has been shown that δ18O is probably more sensitive than SWC for tracing the depth of precipitation infiltration and recharge mechanisms.

I think the paper can be published.

Minor comment:

Formula 1 is not correct, SWC = [(MW - MD)/MW] x 100

Author Response

Response to Reviewer 1 Comments

Point 1: Minor comment: Formula 1 is not correct, SWC = [(MW - MD)/MW] × 100 

Response 1: Thank you very much for your suggestion. Formula 1 has been modified according to your opinion, SWC(%) = [(MW - MD)/MD] × 100. Detailed revised portion are marked in red in revision manuscript.

Response to Reviewer 2 Comments

Point 1: Despite the fact that the authors have done a careful review of the previous studies. Some important references are still missing. I suggest that the authors include the following articles in the revised manuscript for better readership: “Groundwater mixing in the discharge area of San Vittorino Plain (Central Italy): geochemical characterization and implication for drinking uses”. Environmental Earth Sciences, Volume 76, Issue 11, 1 June 2017, Article number 393; “Isotopes in hydrology and hydrogeology” Water (Switzerland) Open Access Volume 11, Issue 2, 7 February 2019, Article number 291.

Response 1: Thank you very much for recommence references. This suggestion was taken. As your opinion, we have added these two references in the introduction: “Barbieri, M.; Nigro, A.; Petitta M., Groundwater mixing in the discharge area of San Vittorino Plain (Central Italy): geochemical characterization and implication for drinking uses. Environmental Earth Sciences 2017, 76, (11), 393.”; “Barbieri, M., Isotopes in Hydrology and Hydrogeology. Water 2019, 11, (2), 291.” Detailed revised portion are marked in red in revision manuscript.

Point 2: In the methods section: what about the Q control of isotopic analyses?

Response 2: Thank you very much for your suggestion. The sample-injection volume (2.50 × 1016 to 4.50 × 1016 water molecules, ± 3.00%), temperature (± 1 °C) and the accuracy of the δ2H and δ18O measurements were checked. The test results that did not satisfy these conditions were eliminated from the analysis.

Point 3: Between the rows 336-342, you cite May precipitation but what about the march isotopic data coming from rain? It’s necessary to add rain from different months.

Response 3: Thank you very much for your suggestion. As your opinion, we have added monthly variation of precipitation amount in the figure 2. Detailed revised portion are marked in red in revision manuscript.

Reviewer 2 Report

Overall, the papers provides a well supported discussion of the role of various land uses on the distribution of oxygen isotopes and soil water.  Content is fine, my issues are minor and are embedded in the attached pdf.

Author Response

Response to Reviewer 2 Comments

Dear reviewer 2,

  Thank you very much for your careful comments and suggestions. This suggestion was taken. As your opinion, we are also extremely greatful to your comments on our manuscript and carefully considered every comment, and made cautious revision accordingly. Detailed revised portion are marked in red in revision manuscript.

Reviewer 3 Report

Dear Editor,

the manuscript is novel and interesting. The manuscript is about the oxygen isotopic composition and the mechanisms of soil water movement in the unsaturated zone in four land use types in the Wangmaogou watershed on the Loess Plateau. Some suggestions:

Despite the fact that the authors have done a careful review of the previous studies. Some important references are still missing. I suggest that the authors include the following articles in the revised manuscript for better readership: “Groundwater mixing in the discharge area of San Vittorino Plain (Central Italy): geochemical characterization and implication for drinking uses”. Environmental Earth Sciences, Volume 76, Issue 11, 1 June 2017, Article number 393; “Isotopes in hydrology and hydrogeology”Water (Switzerland)Open AccessVolume 11, Issue 2, 7 February 2019, Article number 291.

In the methods section: what about the Q control of isotopic analyses?

Tracing soil water movement by cross plotting δ2H and δ18O:

Between the rows 336-342, you cite May precipitation but what about the march isotopic data coming from rain?. It’s necessary to add rain from different months.

Author Response

Response to Reviewer 2 Comments

Point 1: Despite the fact that the authors have done a careful review of the previous studies. Some important references are still missing. I suggest that the authors include the following articles in the revised manuscript for better readership: “Groundwater mixing in the discharge area of San Vittorino Plain (Central Italy): geochemical characterization and implication for drinking uses”. Environmental Earth Sciences, Volume 76, Issue 11, 1 June 2017, Article number 393; “Isotopes in hydrology and hydrogeology” Water (Switzerland) Open Access Volume 11, Issue 2, 7 February 2019, Article number 291.

Response 1: Thank you very much for recommence references. This suggestion was taken. As your opinion, we have added these two references in the introduction: “Barbieri, M.; Nigro, A.; Petitta M., Groundwater mixing in the discharge area of San Vittorino Plain (Central Italy): geochemical characterization and implication for drinking uses. Environmental Earth Sciences 2017, 76, (11), 393.”; “Barbieri, M., Isotopes in Hydrology and Hydrogeology. Water 2019, 11, (2), 291.” Detailed revised portion are marked in red in revision manuscript.

Point 2: In the methods section: what about the Q control of isotopic analyses?

Response 2: Thank you very much for your suggestion. The sample-injection volume (2.50 × 1016 to 4.50 × 1016 water molecules, ± 3.00%), temperature (± 1 °C) and the accuracy of the δ2H and δ18O measurements were checked. The test results that did not satisfy these conditions were eliminated from the analysis.

Point 3: Between the rows 336-342, you cite May precipitation but what about the march isotopic data coming from rain? It’s necessary to add rain from different months.

Response 3: Thank you very much for your suggestion. As your opinion, we have added monthly variation of precipitation amount in the figure 2. Detailed revised portion are marked in red in revision manuscript.

Round  2

Reviewer 1 Report

The authors have addressed all my comments to great extent

 I recommend paper publication

Author Response

Response to Reviewer 1 Comments

Dear reviewer 1,

  Thank you very much for your careful comments and suggestions.

Reviewer 3 Report

The authors followed the reviewers' comments, and now the manuscript is significantly improved and appropriate for publication. My decision is to accept in current form.

Author Response

Response to Reviewer 3 Comments

Dear reviewer 3,

  Thank you very much for your careful comments and suggestions.
